# Defending Against Neural Fake News

**Rowan Zellers♠, Ari Holtzman♠, Hannah Rashkin♠, Yonatan Bisk♠**
**Ali Farhadi♠♡, Franziska Roesner♠, Yejin Choi♠♡**
♠Paul G. Allen School of Computer Science & Engineering, University of Washington
♡Allen Institute for Artificial Intelligence
https://rowanzellers.com/grover

## Abstract

Recent progress in natural language generation has raised dual-use concerns. While applications like summarization and translation are positive, the underlying technology also might enable adversaries to generate *neural fake news*: targeted propaganda that closely mimics the style of real news.

Modern computer security relies on careful *threat modeling*: identifying potential threats and vulnerabilities from an adversary's point of view, and exploring potential mitigations to these threats. Likewise, developing robust defenses against neural fake news requires us first to carefully investigate and characterize the risks of these models. We thus present a model for controllable text generation called GROVER. Given a headline like 'Link Found Between Vaccines and Autism,' GROVER can generate the rest of the article; humans find these generations to be more trustworthy than human-written disinformation.

Developing robust verification techniques against generators like GROVER is critical. We find that best current discriminators can classify neural fake news from real, human-written, news with 73% accuracy, assuming access to a moderate level of training data. Counterintuitively, the best defense against GROVER turns out to be GROVER itself, with 92% accuracy, demonstrating the importance of public release of strong generators. We investigate these results further, showing that exposure bias – and sampling strategies that alleviate its effects – both leave artifacts that similar discriminators can pick up on. We conclude by discussing ethical issues regarding the technology, and plan to release GROVER publicly, helping pave the way for better detection of neural fake news.

## 1 Introduction

Online fake news – news designed to intentionally deceive – has recently emerged as a major societal problem. Malicious actors spread fallacious viral stories in order to gain advertising revenue, influence opinions, and even tip elections (Faris et al., 2017; Wardle and Derakhshan, 2017). As such, countering the spread of disinformation online presents an urgent technical and political issue.

To the best of our knowledge, most disinformation online today is manually written (Vargo et al., 2018). However, as progress continues in natural language generation, malicious actors will increasingly be

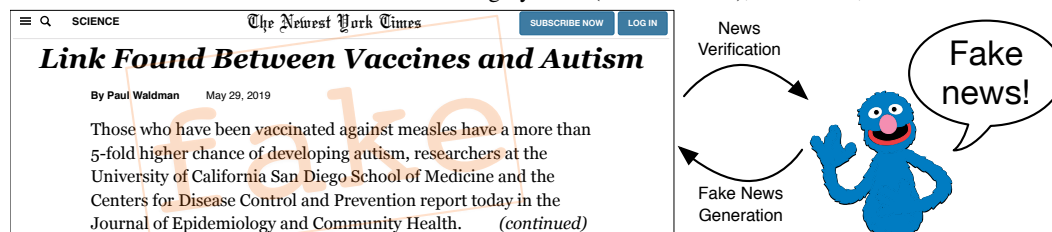

Figure 1: In this paper, we explore GROVER, a model which can detect *and generate* neural fake news. Humans find the articles difficult to distinguish from "real news" without high levels of scrutiny.

able to controllably generate realistic-looking propaganda at scale. Thus, while we are excited about recent progress in text generation (Józefowicz et al., 2016; Radford et al., 2018; 2019), we are also concerned with the inevitability of AI-generated 'neural' fake news.[1]

With this paper, we seek to understand and respond to neural fake news *before* it manifests at scale. We draw on the field of computer security, which relies on *threat modeling*: analyzing the space of potential threats and vulnerabilities in a system to develop robust defenses. To scientifically study the risks of neural disinformation, we present a new generative model called GROVER.[2] Our model allows for controllable yet efficient generation of an entire news article – not just the body, but also the title, news source, publication date, and author list. This lets us study an adversary with controllable generations (e.g. Figure 1, an example anti-vaccine article written in the style of the New York Times).

Humans rate the disinformation generated by GROVER as trustworthy, even more so than human-written disinformation. Thus, developing robust verification techniques against generators such as GROVER is an important research area. We consider a setting in which a discriminator has access to 5000 GROVER generations, but unlimited access to real news. In this setting, the best existing fake news discriminators are, themselves, deep pretrained language models (73% accuracy) (Peters et al., 2018; Radford et al., 2018; 2019; Devlin et al., 2018). However, we find that GROVER, when used in a discriminative setting, performs even better at 92% accuracy. This finding represents an exciting opportunity for defense against neural fake news: the best models for generating neural disinformation are also the best models at detecting it.

Next, we investigate how deep pretrained language models distinguish between real and machine-generated text. We find that key artifacts are introduced during generation as a result of exposure bias: the generator is not perfect, so randomly sampling from its distribution results in generations that fall increasingly out-of-distribution as length increases. However, sampling strategies that alleviate these effects also introduce artifacts that strong discriminators can pick up on.

We conclude with a sketch of the ethical territory that must be mapped out in order to understand our responsibilities as researchers when studying fake news, and the potential negative implications of releasing models (Hecht et al., 2018; Zellers, 2019; Solaiman et al., 2019). Accordingly, we suggest a provisional policy of how such models should be released and why we believe it to be safe – and perhaps even imperative – to do so. We believe our proposed framework and accompanying models provide a concrete initial proposal for an evolving conversation about ML-based disinformation threats and how they can be countered.

## 2   Fake News in a Neural and Adversarial Setting

We present a framework – motivated by today's dynamics of manually created fake news – for understanding what *adversaries* will attempt with deep models, and how *verifiers* should respond.

**Scope of fake news.**   There are many types of *false* news, ranging from satire to propaganda (Wardle, 2017). In this paper, we focus on text-only documents formatted as news articles: stories and their corresponding metadata that contain purposefully false information. Existing fake news is predominantly human-written, for two broad goals: monetization (ad revenue through clicks) and propaganda (communicating targeted information) (Bradshaw and Howard, 2017; Melford and Fagan, 2019). Achieving either goal requires the adversary to be selective about the news that they make, whether by producing only viral content, or content that advances a given agenda.

**Fact checking and verification: related work.**   There is considerable interest in fighting online disinformation. Major platforms such as Facebook prioritize trustworthy sources and shut down accounts linked to disinformation (Mosseri, 2018; Dwoskin and Romm, 2018). Some users of these platforms avoid fake news with tools such as NewsGuard and Hoaxy (Shao et al., 2016) and websites like Snopes and PolitiFact. These services rely on manual fact-checking efforts: verifying the accuracy of claims, articles, and entire websites. Efforts to automate fake news detection generally point out stylistic biases that exist in the text (Rashkin et al., 2017; Wang, 2017; Pérez-Rosas et al.,

2018). These efforts can help moderators on social media platforms shut down suspicious accounts. However, fact checking is not a panacea – cognitive biases such as the backfire effect and confirmation bias make humans liable to believe fake news that fits their worldview (Swire et al., 2017).

**Framework.** We cast fake news generation and detection as an adversarial game, with two players:

- **Adversary**. Their goal is to generate fake stories that match specified attributes: generally, being viral or persuasive. The stories must read realistically to both human users as well as the verifier.
- **Verifier**. Their goal is to classify news stories as real or fake. The verifier has access to unlimited real news stories, but few fake news stories from a specific adversary. This setup matches the existing landscape: when a platform blocks an account or website, their disinformative stories provide training for the verifier; but it is difficult to collect fake news from newly-created accounts.

The dual objectives of these two players suggest an escalating "arms race" between attackers and defenders. As verification systems get better, so too will adversaries. We must therefore be prepared to deal with ever-stronger adversarial attacks, which is the focus of the next section.

## 3   GROVER: Modeling Conditional Generation of Neural Fake News

Given existing online disinformation, we have reason to believe adversaries will try to generate targeted content (e.g. clickbait and propaganda). Recently introduced large-scale generative models produce realistic-looking text (Radford et al., 2019), but they do not lend themselves to producing controllable generations (Hu et al., 2017).[3] Therefore, to probe the feasibility of realistic-looking neural fake news, we introduce GROVER, which produces both realistic *and* controlled generations.

The current state-of-the-art in unconditional text generation views it as a language modeling problem (Bengio et al., 2003), in which the probability of a document $x$ is the product of the conditional probability of generating each token $x_i$ given previous tokens:

$$p(\boldsymbol{x}) = \prod_{i=1}^{N} p(x_i | x_1 \ldots x_{i-1}).  \tag{1}$$

The document is typically treated as a single unstructured *text field*, beginning with a `<start>` token and ending with an `<end>` token. The latter, `<end>`, is particularly important because it indicates the end of the field, and when to should stop generating. However, a news article has necessary structure beyond the running text, or body field. Metadata fields include the domain where the article is published (indirectly marking the style), the date of publication, the names of the authors, and the headline of the article itself. Not only does generating a news article require producing all of these components, these fields also allow significant control over the generations (e.g. specifying a headline helps control the generated body). An article can be modeled by the joint distribution:

$$p(\text{domain}, \text{date}, \text{authors}, \text{headline}, \text{body}).  \tag{2}$$

However, it is not immediately obvious how to sample from Equation 2. One option is to define a *canonical order* among the article's fields $\mathcal{F}$: $(f_1 < f_2 < \ldots < f_{|\mathcal{F}|})$, and model the article left-to-right in that order using Equation 1: $x_1^{f_1}, x_2^{f_1}, \ldots, x_{|f_{|\mathcal{F}|}|}^{f_{|\mathcal{F}|}}$. However, this ordering would forbid sampling certain fields without prohibitively expensive marginalization. Alternatively, one could generate fields in any order, but this requires the model to learn to handle $|\mathcal{F}|!$ potential orderings during inference time.

Our solution is GROVER, a new approach for efficient learning and generation of multi-field documents. We adopt the language modeling framework of Equation 1 in a way that allows for flexible decomposition of Equation 2. During inference time, we start with a set of fields $\mathcal{F}$ as context, with each field $f$ containing field-specific start and end tokens. We sort the fields using a standard order[4] and combine the resulting tokens together. To generate a target field $\tau$, we append the field-specific start token `<start-`$\tau$`>` to the context tokens; then, we sample from the model until we hit `<end-`$\tau$`>`.

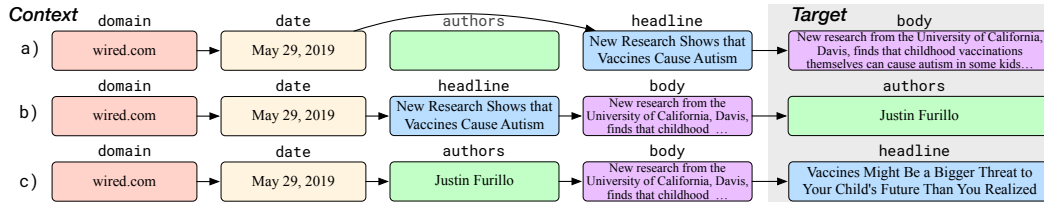

Figure 2: A diagram of three GROVER examples for article generation. In row a), the body is generated from partial context (the authors field is missing). In b), the model generates the authors. In c), the model uses the new generations to regenerate the provided headline to one that is more realistic.

Figure 2 shows an example of using GROVER to generate an anti-vaccine article. Here, the adversary specifies a domain, date, and headline. After GROVER generates the body, it can be used to generate a fake author, before finally generating a new and more appropriate headline.

During training, we simulate inference by randomly partitioning an article's fields into two disjoint sets $\mathcal{F}_1$ and $\mathcal{F}_2$. We also randomly drop out individual fields with probability 10%, and drop out all but the body with probability 35%. This allows the model to learn how to perform unconditional generation. We sort the metadata fields in each set using our standard order, and concatenate the underlying tokens. The model is then trained to minimize the cross-entropy of predicting the tokens in $\mathcal{F}_1$ followed by the tokens in $\mathcal{F}_2$.[5]

**Architecture.** We draw on recent progress in training large Transformers for language modeling (Vaswani et al., 2017), building GROVER using the same architecture as for GPT2 (Radford et al., 2019). We consider three model sizes. Our smallest model, GROVER-Base, has 12 layers and 124 million parameters, on par with GPT and BERT-Base (Radford et al., 2018; Devlin et al., 2018). Our next model, GROVER-Large, has 24 layers and 355 million parameters, on par with BERT-Large. Our largest model, GROVER-Mega, has 48 layers and 1.5 billion parameters, on par with GPT2.

**Dataset.** We present REALNEWS, a large corpus of news articles from Common Crawl. Training GROVER requires a large corpus of news articles with metadata, but none currently exists. Thus, we construct one by scraping dumps from Common Crawl, limiting ourselves to the 5000 news domains indexed by Google News. We used the Newspaper Python library to extract the body and metadata from each article. News from Common Crawl dumps from December 2016 through March 2019 were used as training data; articles published in April 2019 from the April 2019 dump were used for evaluation. After deduplication, REALNEWS is 120 gigabytes without compression.

**Learning.** We trained each GROVER model on randomly-sampled sequences from REALNEWS with length 1024. Other optimization hyperparameters are in Appendix A. We trained GROVER-Mega for 800k iterations, using a batch size of 512 and 256 TPU v3 cores. Training time was two weeks.

## 3.1 Language Modeling results: measuring the importance of data, context, and size

We validate GROVER, versus standard unconditional language models, on the April 2019 test set. We consider two evaluation modes: *unconditional*, where no context is provided and the model must generate the article body; and *conditional*, in which the full metadata is provided as context. In both cases, we calculate the perplexity only over the article body.

Our results, shown in Figure 3, show several conclusions. First, GROVER noticeably improves (between .6 to .9 perplexity points) when conditioned on metadata. Second, perplexity decreases with size, with GROVER-Mega obtaining 8.7 perplexity in the conditional setting. Third, the data distribution is still important: though the GPT2 models with 124M parameters and 355M parameters respectively match our GROVER-Base and GROVER-Large architectures, our model is over 5 perplexity points lower in both cases, possibly because the OpenAI WebText corpus also contains non-news articles.

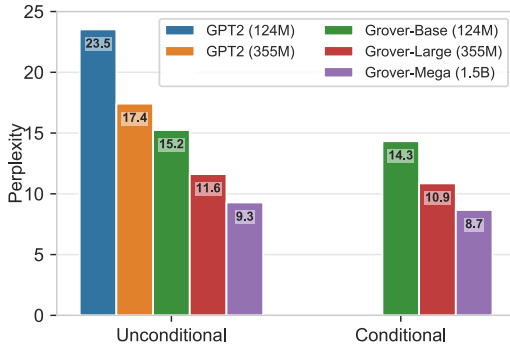

Figure 3: Language Modeling results on the body field of April 2019 articles. We evaluate in the *Unconditional* setting (without provided metadata) as well as in the *Conditional* setting (with all metadata). GROVER sees over a 0.6 point drop in perplexity when given metadata.

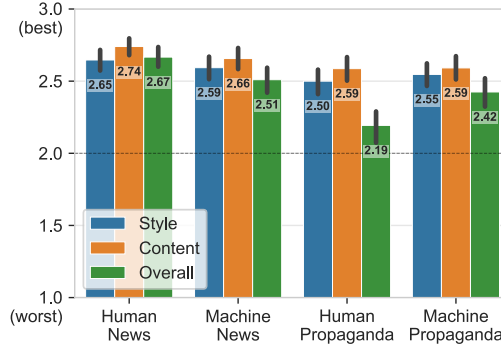

Figure 4: Human evaluation. For each article, three annotators evaluated style, content, and the overall trustworthiness; 100 articles of each category were used. The results show that propaganda generated by GROVER is rated more plausible than the original human-written propaganda.

## 3.2 Carefully restricting the variance of generations with Nucleus Sampling

Sampling from GROVER is straightforward as it behaves like a left-to-right language model during decoding. However, the choice of decoding algorithm is important. While likelihood-maximization strategies such as beam search work well for *closed-ended* generation tasks where the output contains the same information as the context (like machine translation), these approaches have been shown to produce degenerate text during *open-ended* generation (Hashimoto et al., 2019; Holtzman et al., 2019). However, as we will show in Section 6, restricting the variance of generations is also crucial.

In this paper, we primarily use Nucleus Sampling (top-$p$): for a given threshold $p$, at each timestep we sample from the most probable words whose cumulative probability comprises the top-$p$% of the entire vocabulary (Holtzman et al., 2019).[6]

# 4   Humans are Easily Fooled by GROVER-written Propaganda

We evaluate the quality of disinformation generated by our largest model, GROVER-Mega, using $p=.96$. We consider four classes of articles: human-written articles from reputable news websites (`Human News`), GROVER-written articles conditioned on the same metadata (`Machine News`), human-written articles from known *propaganda* websites (`Human Propaganda`), and GROVER-written articles conditioned on the propaganda metadata (`Machine Propaganda`).[7] The domains used are in Appendix B; examples are in Appendix F. We asked a pool of qualified workers on Amazon Mechanical Turk to rate each article on three dimensions: stylistic consistency, content sensibility, and overall trustworthiness.[8]

Results (Figure 4) show a striking trend: though the quality of GROVER-written news is not as high as human-written news, it is adept at rewriting propaganda. The overall trustworthiness score of propaganda increases from 2.19 to 2.42 (out of 3) when rewritten by GROVER.[9]

# 5 Neural Fake News Detection

The high quality of neural fake news written by GROVER, as judged by humans, makes automatic neural fake news detection an important research area. Using models (below) for the role of the *Verifier* can mitigate the harm of neural fake news by classifying articles as `Human` or `Machine` written. These decisions can assist content moderators and end users in identifying likely (neural) disinformation.

**a**. GROVER. We consider a version of our model adapted for discrimination. Similar to GPT (Radford et al., 2018), we place a special `[CLS]` token at the end of each article, and extract the final hidden state at that point. The hidden state is fed to a linear layer to predict the label `Human` or `Machine`.

   To simulate real conditions, and ensure minimal overlap between the generator and discriminator parameters, we initialize GROVER for discrimination using the checkpoint at iteration 700k, whereas the generator uses the checkpoint at iteration 800k.

**b**. GPT2, a 124M or 355M parameter pretrained Transformer language model. Similar to GROVER, we follow the GPT approach and extract the hidden state from a newly-added `[CLS]` token.

**c**. BERT, a 110M parameter (BERT-Base) or 340M parameter (BERT-Large) bidirectional Transformer encoder commonly used for discriminative tasks. We perform domain adaptation to adapt BERT to the news domain, as well as to account for long articles; details in Appendix C.

**d**. FastText, an off-the-shelf library for bag-of-ngram text classification (Joulin et al., 2017). Though not pretrained, similar models do well at detecting human-written fake news.

All models are trained to minimize the cross-entropy loss of predicting the right label. Hyperparameters used during discrimination are in Appendix D.

## 5.1 A semi-supervised setting for neural fake news detection

While there are many human-written articles online, most are from the distant past, whereas articles to be detected will likely be set in the present. Likewise, there might be relatively few neural fake news articles from a given adversary.[10] We thus frame neural fake news detection as a semi-supervised problem. A neural verifier (or *discriminator*) has access to many human-written news articles from March 2019 and before – the entire REALNEWS training set. However, it has limited access to generations, and more recent news articles. Using 10k news articles from April 2019, we generate article body text; another 10k articles are used as a set of human-written news articles. We split the articles in a balanced way, with 10k for training (5k per label), 2k for validation, and 8k for testing.

We consider two evaluation modes. In the **unpaired** setting, a discriminator is provided single news articles, and must classify each independently as `Human` or `Machine`. In the **paired** setting, a model is given two news articles with the same metadata, one real and one machine-generated. The discriminator must assign the machine-written article a higher `Machine` probability than the human-written article. We evaluate both modes in terms of accuracy.

## 5.2 Discrimination results: GROVER performs best at detecting GROVER's fake news

We present experimental results in Table 1 for all generator and discriminator combinations. For each pair, we show the test results using the most adversarial generation hyperparameters (top-$p$) as judged on the validation set.[11] The results show several trends. First, the paired setting appears much easier than the unpaired setting, suggesting that it is difficult for the model to calibrate its predictions. Second, model size is highly important in the arms race between generators and discriminators. Using GROVER to discriminate GROVER's generations results in roughly 90% accuracy across the range of sizes. If a larger generator is used, accuracy slips below 81%; conversely, if the discriminator is larger, accuracy is above 98%. Third, other discriminators perform worse than GROVER overall, even when controlling for architecture size and (for both BERT models) the domain.

That GROVER is the best discriminator is possibly surprising: being unidirectional, it is less expressive than deep bidirectional models such as BERT.[12] That the more expressive model here is **not** the best at

Table 1: Results of discriminators versus generators, in both the paired and unpaired settings and across architecture sizes. We also vary the generation hyperparameters for each generator-discriminator pair, reporting the discrimination test accuracy for the hyperparameters with the *lowest* validation accuracy. Compared with other models such as BERT, GROVER is the best at detecting its own generations as neural fake news.

|  |  | Unpaired Accuracy Generator size | | | Paired Accuracy Generator size | | |
|---|---|---|---|---|---|---|---|
|  |  | 1.5B | 355M | 124M | 1.5B | 355M | 124M |
|  | Chance | | 50.0 | | | 50.0 | |
| 1.5B | GROVER-Mega | **92.0** | **98.5** | **99.8** | **97.4** | **100.0** | **100.0** |
| 355M | GROVER-Large | **80.8** | **91.2** | **98.4** | **89.0** | **96.9** | **100.0** |
|  | BERT-Large | 73.1 | 75.9 | 97.5 | 84.1 | 91.5 | 99.9 |
|  | GPT2 | 70.1 | 78.0 | 90.3 | 78.8 | 87.0 | 96.8 |
| 124M | GROVER-Base | **70.1** | **80.0** | **89.2** | 77.5 | 88.2 | 95.7 |
|  | BERT-Base | 67.2 | 76.6 | 84.1 | **80.0** | **89.5** | **96.2** |
|  | GPT2 | 66.2 | 71.9 | 83.5 | 72.5 | 79.6 | 89.6 |
| 11M | FastText | 63.8 | 65.6 | 69.7 | 65.9 | 69.0 | 74.4 |

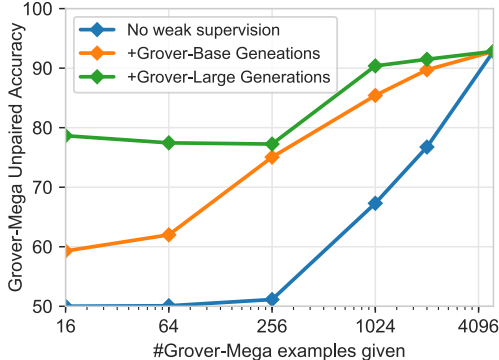

Figure 5: Exploring weak supervision for discriminating GROVER-Mega generations. With no weak supervision, the discriminator sees $x$ machine-written articles (from GROVER Mega). For +GROVER-Base and +GROVER-Mega, the discriminator sees $5000-x$ machine-written articles given by the weaker generator in question. Seeing weaker generations improves performance when few in-domain samples are given.

discriminating between real and generated news articles suggests that neural fake news discrimination requires having a similar *inductive bias* as the generator.[13]

### 5.3 Weak supervision: what happens if we don't have access to GROVER-Mega?

These results suggest that GROVER is an effective discriminator when we have a medium number of fake news examples from the exact adversary that we will encounter at test time. What happens if we relax this assumption? Here, we consider the problem of detecting an adversary who is generating news with GROVER-Mega and an unknown top-$p$ threshold.[14] In this setup, during training, we have access to a weaker model (GROVER-Base or GROVER-Large). We consider the effect of having only $x$ examples from GROVER-Mega, and sampling the missing $5000-x$ articles from one of the weaker models, where the top-p threshold is uniformly chosen for each article in the range of $[0.9, 1.0]$.

We show the results of this experiment in Figure 5. The results suggest that observing additional generations greatly helps discrimination performance when few examples of GROVER-Mega are available: weak supervision with between 16 and 256 examples from GROVER-Large yields around 78% accuracy, while accuracy remains around 50% without weak supervision. As the portion of examples that come from GROVER-Mega increases, however, accuracy converges to 92%.[15]

## 6 How does a model distinguish between human and machine text?

In this section, we explore why GROVER performs best at detecting fake news generated by other GROVER models. We find that there is a double-bind between **exposure bias** and **variance-reduction** algorithms that alleviate these biases while at the same time creating other artifacts.

**Exposure Bias.** Models maximizing Equation 1 are trained only conditioned on human-written text, never on its own generations, creating a problem known as exposure bias (Ranzato et al., 2016).

We investigate the importance of exposure bias towards creating artifacts. In Figure 6 we plot the perplexities given by GROVER-Mega over each position for body text at top-$p$ thresholds of 0.96 and 1, as well as over human text. Generating the first token after <startbody> results in high

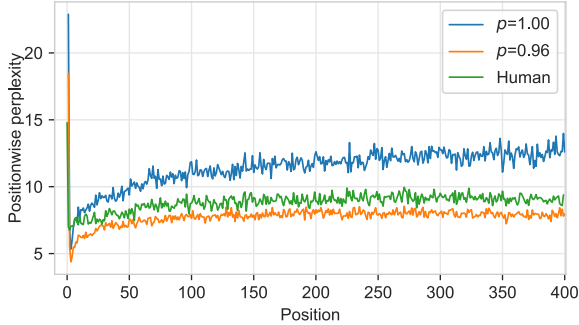

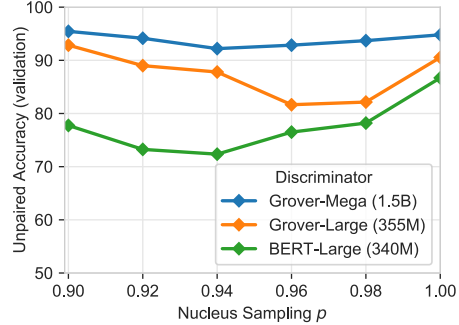

Figure 6: Perplexities of Grover-Mega, averaged over each position in the body (after conditioning on metadata). We consider human-written with Grover-Mega generated text at $p$=1 (random sampling) and $p$=.96. The perplexity of randomly sampled text is higher than human-written text, and the gap increases with position. This suggests that sampling without variance reduction increasingly falls out-of-distribution.

Figure 7: Unpaired validation accuracy, telling apart generated news articles (from Grover Mega) from real articles, at different variance reduction thresholds $p$ (for Nucleus Sampling). Results varying $p$ show a sweet spot ($p$ = 0.92 – 0.96) wherein discrimination is hardest.

perplexity. However, the rest of the positions show a curious pattern: the perplexity of human-written text is lower than randomly sampled text, and this gap increases with sequence length, suggesting that random sampling causes Grover to fall increasingly out of the distribution of human language. However, limiting the variance ($p$=0.96) lowers the resulting perplexity and limits its growth.

**Limiting the variance of a model also creates artifacts**   On the other hand, clipping the model's variance also leaves an artifact, as prior work has observed for top-$k$ sampling (Strobelt and Gehrmann, 2019). A similar phenomenon holds for Nucleus (top-$p$) sampling. The probability of observing a human-written article where all tokens are drawn from the top-$p$% of the distribution is $p^n$, where $n$ is the document's length. This probability goes to zero as $n$ increases. However, for Nucleus Sampled text – in which the final $1-p$ is cut off – all tokens come from the top-$p$.

The visibility of the artifacts depends on the choice of discriminator. The top-$p$ at each timestep is calculated under the generator's worldview, meaning that if the discriminator models text in a different way, it might have a harder time pinpointing the empty $1-p$ tail. This could explain BERT's lower performance during discrimination.

**A sweet spot of careful variance reduction**   Not reducing the variance, as well as significantly reducing the variance, both cause problems. Might there be a *sweet spot* for how much to truncate the variance, to make discrimination maximally hard? In Figure 7, we show results varying the top-$p$ threshold for the discrimination task applied to Grover-Mega's generations. The results indeed show a sweet spot, roughly between $p$=0.92 and $p$=0.98 depending on the discriminator, wherein discrimination is hardest. Interestingly, we note that the most adversarial top-$p$ threshold for BERT-Large is considerably lower than the corresponding top-$p$ for Grover-Large of the same size. This supports our hypothesis that BERT's view of language differs markedly from Grover; using a lower top-$p$ threshold does not seem to give it much more information about the missing tail.

**Overall**, our analysis suggests that Grover might be the best at catching Grover because it is the best at knowing where the tail is, and thus whether it was truncated.

# 7   Conclusion: a Release Strategy for Grover

This paper investigates the threats posed by adversaries seeking to spread disinformation. Our sketch of what these threats might look like – a controllable language model named Grover – suggests that these threats are real and dangerous. Grover can rewrite propaganda articles, with humans rating the rewritten versions as more trustworthy. At the same time, there are defenses to these models – notably, in the form of Grover itself. We conclude with a discussion of next steps and ethical considerations.

**The Era of Neural Disinformation.** Though training GROVER was challenging, it is easily achievable by real-world adversaries today. Obtaining the data required through Common Crawl cost $10k in AWS credits and can be massively parallelized over many CPUs. Training GROVER-Mega is relatively inexpensive: at a cost of $0.30 per TPU v3 core-hour and two weeks of training, the total cost is $25k. Spending more money and engineering time could yield even more powerful generators.

**Release of generators is critical.** At first, it would seem like keeping models like GROVER private would make us safer. However, GROVER serves as an effective detector of neural fake news, even when the generator is much larger (Section 5). If generators are kept private, then there will be little recourse against adversarial attacks. We thus released our models to researchers (Zellers, 2019).

**Future of progress in generation.** Models like BERT are strong discriminators for many NLP tasks, but they are not as good at detecting GROVER's generations as left-to-right models like GROVER, even after domain adaptation. One hypothesis is that the artifacts shown in Section 6 are most visible to a left-to-right discriminator. This also suggests that recent progress on generating text in any order (Gu et al., 2019; Stern et al., 2019; Ghazvininejad et al., 2019) may lead to models that evade a GROVER discriminator. Likewise, models that are trained conditioned on their own predictions might avoid exposure bias, however, these objectives often lead to low performance on language tasks (Caccia et al., 2018). One additional possibility is the use of Adversarial Filtering (Zellers et al., 2018; 2019b) to oversample and then select a subset of generations. However, we found this didn't work well for very long sequences (up to 1024 BPE tokens), possibly as these are far from the 'Goldilocks Zone' wherein discrimination is hard for machines.

**Additional threat models.** In this paper, we studied the threat model whereby an adversary generates an entire news article from scratch, given minimal context. Other threat models are possible: for instance, an adversary might generate comments or have entire dialogue agents, they might start with a human-written news article and modify a few sentences, and they might fabricate images or video. These threat models ought to be studied by researchers also so that we can create better defenses.

**Machine-generated real news?** Our study focused on detecting machine-written fake news, though the same GROVER approach can be used for spotting human-written fake news as well (Zellers et al., 2019c). However, machines can also generate truthful news using templated systems. Domains with templated news articles exist in our dataset,[16] and are easy for GROVER to spoof convincingly.

**Future of progress in discrimination.** Our discriminators are effective, but they primarily leverage distributional features rather than evidence. In contrast, humans assess whether an article is truthful by relying on a model of the world, assessing whether the evidence in the article matches that model. Future work should investigate integrating knowledge into the discriminator (e.g. for claim verification in FEVER; Thorne et al., 2018). An open question is to scale progress in this task towards entire news articles, and without paired evidence (similar to open-domain QA; Chen et al., 2017).

**What should platforms do?** Video-sharing platforms like YouTube use deep neural networks to scan videos while they are uploaded, to filter out content like pornography (Hosseini et al., 2017). We suggest platforms do the same for news articles. An ensemble of deep generative models, such as GROVER, can analyze the content of text – together with more shallow models that predict human-written disinformation. However, humans must still be in the loop due to dangers of flagging real news as machine-generated, and possible unwanted social biases of these models.

## Acknowledgments

We thank the anonymous reviewers, as well as Dan Weld, for their helpful feedback. Thanks also to Zak Stone and the Google Cloud TPU team for help with the computing infrastructure. This work was supported by the National Science Foundation through a Graduate Research Fellowship (DGE-1256082) and NSF grants (IIS-1524371, 1637479, 165205, 1703166), the DARPA CwC program through ARO (W911NF-15-1-0543), the Sloan Research Foundation through a Sloan Fellowship, the Allen Institute for Artificial Intelligence, the NVIDIA Artificial Intelligence Lab, Samsung through a Samsung AI research grant, and gifts by Google and Facebook. Computations on beaker.org were supported in part by credits from Google Cloud.

## Footnotes

[1] We thank past work, such as OpenAI's Staged Release Policy for GPT2 for drawing attention to neural disinformation, alongside other dual-use implications.

[2] Short for **G**enerating a**R**ticles by **O**nly **V**iewing m**E**tadata **R**ecords.

[3] A common workaround is to have a human seed the text to provide context. However, this **a)** is a heavy handed technique for biasing which may not capture the desired attributes, and **b)** leaves in place a human-written beginning (as tokens are only generated left-to-right), which may create distributional artifacts.

[4] Our ordering is the following field types in order: domain, date, authors, headline, and then the body.

[5]All tokens use the same vocabulary. By using a standard order, but partitioning the fields into two sets, the model can generate any field conditioned on others while only needing to learn $2^{|\mathcal{F}|}$ orderings, versus $|\mathcal{F}|!$.

[6]In early experiments, we found Nucleus Sampling produced better and less-detectable generations than alternatives like top-$k$ sampling, wherein the most probable $k$ tokens are used at each timestep (Fan et al., 2018).

[7]We use the technique described in Figure 2 to rewrite the propaganda: given the metadata, generate the article first, and then rewrite the headline.

[8]With these guidelines, we tried to separate style versus content. Overall trustworthiness asks 'Does the article read like it comes from a trustworthy source?' which emphasizes style, while content sensibility asks whether the content is believable on a semantic level.

[9]This difference is statistically significant at $p = 0.01$. One possible hypothesis for this effect is that GROVER ignores the provided context. To test this hypothesis, we did a human evaluation of the consistency of the article body with the headline, date, and author. We found that human-written propaganda articles are consistent with the headline with an average score of 2.85 of 3 on the same 1-3 scale, while machine-written propaganda is consistent with 2.64 of 3.

[10]Moreover, since disinformation can be shared on a heterogeneous mix of platforms, it might be challenging to pin down a single generated model.

[11]For each discriminator/generator pair, we search over $p \in \{.9, .92, .94, .96, .98, 1.0\}$.

[12]Indeed, bidirectional approaches perform best on leaderboards like GLUE (Wang et al., 2018).

[13]This matches findings on the HellaSwag dataset (Zellers et al., 2019b). Given human text and machine text written by a finetuned GPT model, a GPT discriminator outperforms BERT-Base at picking out human text.

[14]The top-$p$ threshold used was $p=0.96$, but we are not supposed to know this!

[15]In additional experiments we show that accuracy increases even more – up to 98% – when the number of examples is increased (Zellers et al., 2019c). We also find that GROVER when trained to discriminate between real and fake GROVER-generated news can detect GPT2-Mega generated news as fake with 96% accuracy.

[16]An example is https://americanbankingnews.com.

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
