[Supplementary Material]

## Supplemental Material

## A Optimization Hyperparameters

For our input representation, we use the same BPE vocabulary as (Radford et al., 2019). We use Adafactor (Shazeer and Stern, 2018) as our optimizer. Common optimizers such as Adam (Kingma and Ba, 2014) tend to work well, but the memory cost scales linearly with the number of parameters, which renders training GROVER-Mega all but impossible. Adafactor alleviates this problem by factoring the second-order momentum parameters into a tensor product of two vectors. We used a maximum learning rate of 1e-4 with linear warm-up over the first 10,000 iterations, and decay over the remaining iterations. We set Adafactor's $\beta_1 = 0.999$ and clipped updates for each parameter to a root-mean-squared of at most 1. Last, we applied weight decay with coefficient 0.01. We used a batch size of 512 on 256 TPU v3 cores. which corresponds to roughly 20 epochs through our news dataset. The total training time required roughly two weeks.

## B Real News and Propaganda Websites

In our generation experiments (Section 4), we consider a set of mainstream as well as propaganda websites. We used the following websites as 'real news': `theguardian.com`, `reuters.com`, `nytimes.com`, `theatlantic.com`, `usatoday.com`, `huffingtonpost.com`, and `nbcnews.com`. For propaganda sites, we chose sites that have notably spread misinformation (Dicker, 2016) or propaganda[17]. These were `breitbart.com`, `infowars.com`, `wnd.com`, `bigleaguepolitics.com`, and `naturalnews.com`.

## C Domain Adaptation of BERT

BERT (Devlin et al., 2018) is a strong model for most classification tasks. However, care must be taken to format the input in the right way, particularly because BERT is pretrained in a setting where it is given two spans (separated by a special `[SEP]` token). We thus use the following input format. The first span consists of the metadata, with each field prefixed by its name in brackets (e.g. '`[title]`'). The second span consists of the body. Because the generations are cased (with capital and lowercase letters), we used the 'cased' version of BERT.

Past work (e.g. Zellers et al. (2019a); Han and Eisenstein (2019)) has found that BERT, like other language models, benefits greatly from domain adaptation. We thus perform domain adaptation on BERT, adapting it to the news domain, by training it on REALNEWS for 50k iterations at a batch size of 256. Additionally, BERT was trained with a sequence length of at most 512 WordPiece tokens, but generations from GROVER are much longer (1024 BPE tokens). Thus, we initialized new position embeddings for positions 513-1024, and performed domain adaptation at a length of 1024 WordPiece tokens.

## D Hyperparameters for the Discriminators

For our discrimination experiments, we limited the lengths of generations (and human-written articles) to 1024 BPE tokens. This was needed because our discriminators only handle documents up to 1024 words. However, we also found that the longer length empirically discrimination easier for models (see Section 6).

For our discrimination experiments, we used different hyperparameters depending on the model, after an initial grid search. For BERT, we used the Adam (Kingma and Ba, 2014) optimizer with a learning rate of $2e-5$ and a batch size of 64. We trained BERT models for 5 epochs, with a linear warm-up of the learning rate over the initial 20% iterations. For GPT2 and GROVER, we used the Adam actor optimizer (Shazeer and Stern, 2018) optimizer with a learning rate of $2e-5$ for all models, and a batch size of 64. We applied an auxiliary language modeling loss for these models with a coefficient of 0.5. These models were trained for 10 epochs, with a linear warm-up over the initial 20% iterations.

# E   Human Evaluation Prompt

## E.1   Evaluating Quality

For evaluating the quality of GROVER-written versus human-written news articles, we asked workers the following questions (shown exactly). The answer choices are shown next to the rating under our 1-3 Likert scale (3 being the best, 1 being the worst for each attribute).

(a) (Style) Is the style of this article consistent?

  3. **Yes**, this sounds like an article I would find at an online news source.
  2. **Sort of**, but there are certain sentences that are awkward or strange.
  1. **No**, it reads like it's written by a madman.

(b) (Content) Does the content of this article make sense?

  3. **Yes**, this article reads coherently.
  2. **Sort of**, but I don't understand what the author means in certain places.
  1. **No**, I have no (or almost no) idea what the author is trying to say.

(c) (Overall) Does the article read like it comes from a trustworthy source?

  3. **Yes**, I feel that this article could come from a news source I would trust.
  2. **Sort of**, but something seems a bit fishy.
  1. **No**, this seems like it comes from an unreliable source.

## E.2   Evaluating consistency

To measure consistency between the article and the metadata, we asked the following questions:

(a) (Headline) How well does the article body match the following headline? [headline]

  3. **Yes**, the article makes sense as something that I would see given the headline.
  2. **Sort of**, the article is somewhat related to the headline, but seems slightly off.
  1. **No**, the article is completely off-topic.

(b) (Authors) How well does the article body match the following author(s)? [authors]

  3. **Yes**, the article makes sense as something that could be written by the author(s).
  2. **Sort of**, the article might have been written by the author(s) above, but it sounds unlikely.
  1. **No**, the article body contains information that says it was written by someone else.

(c) (Date) How well does the article body match the following date? [date]

  3. **Yes**, the article makes sense as something that could have been written on [date].
  2. **Sort of**, the article might have been written on [date], but it sounds unlikely.
  1. **No**, there's information in the article that conflicts the proposed date.

# F   Examples

In Figures 8 and 9, we include examples of articles with the average scores given by human raters, who were asked to evaluate the style, content, and overall trustworthiness. In Figure 8, we show a real article (`Human News`) posted by the Guardian along with an article from GROVER (`Machine News`) made using the same metadata. Figure 9 shows a real propaganda article from the Natural News (`Human Propaganda`) and an article made with GROVER (`Machine Propaganda`) with the original headline and the style of Huffington Post (GROVER was used to re-write the title to be more stylistically similar to the Huffington Post, as well).

We also present several other generated examples, generated from GROVER-Mega with a top-$p$ threshold of $p$=0.95. All of the examples are cut off to 1024 generated BPE tokens, since this is our setup for discrimination.

**a**. GROVER can generate controlled propaganda. In Figure 10, we show the continuation from Figure 1, about a link found between autism and vaccines.

**Original Headline: Timing of May's 'festival of Britain' risks Irish anger**

**Human-written News Article**

**Timing of May's 'festival of Britain' risks Irish anger**
April 13, 2019  theguardian.com

It was meant to be a glimmer of positivity to unite a divided nation – a festival to celebrate the best of British, bring communities together and strengthen "our precious union".

Yet Theresa May is being warned that her plan for a Festival of Great Britain and Northern Ireland risks doing the opposite. The planned 2022 event, announced at last year's Conservative conference, was criticised as a headline-grabbing distraction. But May now faces concerns that the timing clashes with the centenary of Irish partition and the civil war. Arts industry figures in Northern Ireland and some of those involved in the peace process are also understood to have concerns. These worries are revealed in a report by the thinktank British Future, which examined the potential for arts and heritage to bring the nation together. The study calls on the festival to be delayed by at least three years.

What is now the Irish republic became the Irish Free State in 1922, while Northern Ireland remained part of the UK. A civil war erupted among Irish nationalists over the remaining links with Britain and raged for a year. Sunder Katwala, the report's author, said: "Holding a festival of Great Britain and Northern Ireland in 2022, on the centenary of Ireland's partition and civil war, would be the worst possible timing. It is only likely to heighten tensions between communities – and that's before we know Brexit's implications for the border. Right across the UK, a festival so closely associated with Brexit may only reinforce divides when it could be bridging them."

Jonathan Powell, Tony Blair's former chief of staff and one of the architects of the Good Friday agreement, also warned against anything that could inflame tensions, which have worsened given the prominent role of the Irish border in the Brexit debate.

"There is going to be a lot of rebuilding to do in Anglo-Irish relations, which had been put on a completely new footing over the last 20 years and were then trashed by the Brexiters over the last couple of years," he said. "An example is their constant attacks on [Irish prime minister Leo] Varadkar for hostility when all he is doing is protecting the interests of the Irish people. I fear it will take years to put this right."

The idea of a nationwide event was modelled on the Great Exhibition of 1851 and the Festival of Britain a century later, held to boost the nation's spirits after the second world war.

May was mocked on social media when she announced she was setting aside £120m to prepare for the event, with some suggesting the festival would only feature damp tents on village greens, an exhibition of the remaining avocados left in Britain and morris dancers. However, after holding focus groups, British Future found it could be more popular than critics think, should organisers get the tone and timing right. More than six in 10 thought the festival a good idea, with only one in 10 opposed, according to new ICM research.

"The festival could be much more popular with the public than its social media critics think, and could help bridge social divides if done well," Katwala said. "If not, it could be a second Millennium Dome. It risks being neglected, with the government stuck over Brexit and the prime minister who thought of it on her way out. The next prime minister will have to decide how to rescue this idea."

**Ratings**
Style: 3.0
Content: 3.0
Overall: 3.0

**Machine-written News Article**

**Timing of May's 'festival of Britain' risks Irish anger**
April 13, 2019  theguardian.com

The prime minister's decision to launch her much-hyped "great Britain" conference on Monday – the same day as a Lisbon treaty event paving the way for Brexit-free member states to leave the European Union – will be seen as provocative by some, according to senior Tories.

Jo Johnson said May's statement in Edinburgh was "instrumental" in chipping away at the strength of domestic opposition to Brexit. He added that the prime minister had also "churned" the membership of the ERG back into service.

Tom Pursglove, another Tory MP involved in the campaign to prevent Brexit, said: "By lifting the gagging order on ERG members from the Liaison Committee and starting an intensification of the ethnic profiling of Remainers, the prime minister is doing herself and the ERG proud."

Announcing that the conference would launch her vision for the country, May will call for more global Britain to fight for global trade. Although still committed to leaving the single market and customs union, the Conservatives want to highlight the importance of these deals – as well as tackling climate change, tackling modern slavery and tackling poverty.

The event will be on Monday 29 April, the day before the EU's 2019 budget is agreed. May's Treasury chief secretary, Liz Truss, is to try to convince European finance ministers that there is no alternative plan to Brexit. EU officials and political leaders are scheduled to decide the EU's £1.2tn budget in mid-October. The Northern Ireland-based DUP, which failed to back May in the no confidence vote she suffered earlier this month, will be encouraged by the event. The DUP said it would be "easy to ignore" the motions at the conference, but would vote against any effort to transfer powers to Brussels.

Labour MP Sir Keir Starmer, who now chairs the cross-party Brexit negotiations committee, said: "The timing of her conference announcement raises some worrying issues. We cannot allow the UK's terms of exit to be dictated by no confidence votes.

"These checks cannot be on the superficial level, where some make noises on the hill but are wholly unwilling to set out detailed proposals. Tighter controls at Heathrow are essential, and if May really wants to celebrate 'all change', then she should close Britain's borders for a week and see how workable it is to stop EU nationals from flying in on the same visa system as Brits.

"Brexit would be fantastic for the business world if you measure economic value only on the quality of the deal. But – and when we say 'if' the prime minister doesn't care that she is still far short of securing that 'good deal' – she needs to work harder to deliver that for her negotiators."

Other critics, including party member James Ball, drew parallels with Brexit minister Dominic Raab's similar focus on trade deals to stop other EU states leaving the bloc. They said Raab's speech last week was "the latest Labour-held ploy to quietly delay Brexit, run out the clock or blame everyone except the UK for not being willing to walk away".

• Follow Guardian Opinion on Twitter at @gdnopinion

**Ratings**
Style: 3.0
Content: 3.0
Overall: 2.3

Figure 8: Example of human-written news and machine-written news articles about the same headline from The Guardian with the average ratings from human rating study.

**b**. GROVER can spoof the identity of writers. In Figure 11 we show a realistic-looking editorial seemingly from New York Times columnist Paul Krugman.

**c**. GROVER can generate fake political news. In Figure 12 we show an article generated about Trump being impeached, written in the style of the Washington Post.

**d**. GROVER can generate fake movie reviews (opinion spam; Ott et al. (2011)). In Figure 13 we show a movie review, generated in the style of LA Times Movie Critic Kenneth Turan, for Sharknado 6, 'The Last Sharknado: It's About Time'

**e**. GROVER can generate fake business news. In Figure 14, we show an article generated about an 'Uber for Dogs' startup.

**Original Headline: Don't drink the water: The dark side of water fluoridation**

**Don't drink the water: The dark side of water fluoridation**
March 13. 2019 naturalnews.com

(Natural News) There are 7.7 billion people on this planet (as at March 2019). Only about 5 percent of them drink fluoridated water. Why? Because their governments recognize that fluoride in large amounts becomes a toxic chemical that is not fit for human consumption. The 328,000,000 citizens of the United States drink more fluoridated water than all other countries combined. Why? Because the U.S. government continues to doggedly insist that it is safe and improves dental health.
But what do the facts say? As reported by Waking Times, dozens of peer-reviewed studies published in prestigious journals like The Lancet, have confirmed that fluoride is in fact toxic – especially to the developing brains of children. These chemicals are derived from unprocessed toxic waste which is not purified in any way before being pumped into the water supply. How could it possibly be anything but harmful?
The history of water fluoridation in the United States
So, what prompted the government to start adding something so obviously harmful to our precious water supply?
Waking Times, quoting from an article by The Children's Health Defense Team, explains a little about the history of this practice:
During World War II, fluoride (a compound formed from the chemical element fluorine) came into large-scale production and use as part of the Manhattan Project. According to declassified government documents summarized by Project Censored, Manhattan Project scientists discovered early on that fluoride was a "leading health hazard to bomb program workers and surrounding communities." In order to stave off lawsuits, government scientists "embarked on a campaign to calm the social panic about fluoride…by promoting its usefulness in preventing tooth decay."
The power of the elements: Discover Colloidal Silver Mouthwash with quality, natural ingredients like Sangre de Drago sap, black walnut hulls, menthol crystals and more. Zero artificial sweeteners, colors or alcohol. Learn more at the Health Ranger Store and help support this news site.
To back up its decision, the government embarked on a series of flawed and poorly designed "scientific" studies, which an expert later lambasted as "especially rich in fallacies, improper design, invalid use of statistical methods, omissions of contrary data, and just plain muddleheadedness and hebetude."
They then used these sham studies to enforce a national policy of water fluoridation.
Studies confirm fluoride lowers IQ and harms children in other ways
Interestingly, even government-backed studies have confirmed the dangers of fluoride in drinking water. For example, a study published in 2017, which was largely funded by the government's National Institutes of Health and National Institute of Environmental Health Sciences, uncovered a "strong relationship" between fluoride exposure in the womb and reduced cognitive function.
In addition, Natural News previously reported:
More than 50 peer-reviewed studies have linked the consumption of fluoridated water to lower IQ in children. A joint metanalysis by Harvard School of Public Health and China Medical University, which examined 27 studies on the subject, found "strong indications that fluoride may adversely affect cognitive development in children."
Fluoridation has also been linked to countless other devastating health effects in children, including premature birth, impaired neurological development, autism and preeclampsia. A recent study also confirmed a significant link between fluoridation and ADHD.
Death rates up to 26 percent higher in the most fluoridated states
And it gets worse. As reported by Waking Times, death rates among people who live in the 10 states with the highest fluoridation levels are between 5 and 26 percent higher than among people living in the 10 least fluoridated states.
Furthermore, studies have linked fluoridated water to cancer, damaging effects on the brain, and dental fluorosis, which experts agree is the first visible sign of fluoride toxicity.
The National Research Council of the National Academies also issued a warning in 2006 that fluoride exposure might be a risk factor in the development of Alzheimer's disease.
And to top it all off, there is literally no evidence that fluoride even prevents cavities! Even if it did, would that small benefit really be worth the massive potential for serious health problems that fluoridation has been scientifically linked to?
Learn more at Fluoride.news or TapWater.news.
Sources include:
WakingTimes.com
NaturalNews.com
ChildrensHealthDefense.org

| Ratings |
|---|
| Style: 2.3 |
| Content: 2.3 |
| Overall: 1.7 |

**Fluoride in Your Water Isn't Healthy, Even When It's Recommended**
March 13, 2019 huffingtonpost.com

Fluoride is regarded by researchers around the world as the "gold standard" in tooth care, and a safe, common ingredient that has almost been universally found to be safe in past studies of health effects. It's absorbed without interference from the body's natural minerals.
Researchers at the U.S. Geological Survey (USGS) published the results of a multi-state environmental health study last month. It showed that during the first three decades of fluoridation of tap water systems, fluoride produced from the process alone increased rates of dental caries (the biggest contributor to tooth decay) by 16 percent in Mississippi and a whopping 45 percent in Arizona, which implemented fluoridation systems back in 1942. This increase was seen after a decade when fluoride levels didn't change.
USGS also found that fluoridation increased rates of other toxicants and petrochemicals, as well as deaths from brain, lung, kidney and bladder cancer.
It bears noting that there is no clear proof that these specific contaminants were caused by fluoridation, but the USGS study at least hints that this was the case. The epidemic of brain cancers across the U.S. — especially in teenagers — has confounded researchers for decades. The USGS study points to links to numerous studies that have linked water fluoridation with increased risks of cancer.
Even though the majority of studies on water fluoridation have not produced such alarming results, the mainstream medical community is, apparently, still skeptical. Two years ago, doctors from Harvard and Duke universities suggested that fluoride is associated with lower IQ scores and autoantibodies to water. The results of a recent study that followed more than 700 children over a period of four years demonstrated that the kids were more likely to have symptoms of illness, more likely to have higher blood pressure and sleep problems, had higher mean energy expenditure, more struggles with attention and poorer concentration and performance on educational tasks. The data also showed that the children were not more vulnerable to fluoride when it was administered by toothpaste.
Even though the majority of studies on water fluoridation have not produced such alarming results, the mainstream medical community is, apparently, still skeptical.
The link between bacteria and tooth decay is legendary. Experts agree that fluoride erases a cavity's effect on surface-level bacteria and increases decay via dangerous pathways to deeper tissue. Fluoride actually quarantines surface-level bacteria to caries-causing toxins in the plaque and oral cavity. The longer the fluoride is in contact with those toxins, the more damage and damage can occur.
Evaluations of the impacts of water fluoridation are well underway, but we've seen too many conflicts of interest and dirty laundry in the past few years. One of the largest of those is the Water Fluoridation Corporation (WFC), an industry network that carries on and even finances the work of paid "expert witnesses" in both domestic and international water fluoridation litigation. The WFC boasts 38 former and current members of the American Dental Association, including board members, former ADA commissioners, and representatives of the ADA-funded American Academy of Oral Medicine and American Dental Hygienists. It is arguably the second largest source of fluoride in the U.S. right behind the ADA.
Despite denials, none of this new water research paints a rosy picture of water fluoridation. We can blame these new findings on ongoing conflicts of interest and blatant biases.
Fluoride in your drinking water isn't safe, even when it's recommended. There are three risks that we as consumers need to ensure that are understood: 1) the hidden contaminants that may be causing harm; 2) the effects of over-medicating the fluoride vaccine; and 3) low-dose levels of fluoride in very old bones, causing unnecessary bone damage and irreparable nerve damage.

| Ratings |
|---|
| Style: 2.3 |
| Content: 2.7 |
| Overall: 2.7 |

Figure 9: Example of human-written and machine-written articles arguing against fluoride with the average ratings from human rating study.

Provided headline: 'Link found between autism and vaccines'

Generated body: Those who have been vaccinated against measles have a more than 5-fold higher chance of developing autism, researchers at the University of California San Diego School of Medicine and the Centers for Disease Control and Prevention report today in the Journal of Epidemiology and Community Health.

Those who have been vaccinated against measles have a more than 5-fold higher chance of developing autism. Earlier studies had linked past vaccines with the condition, but it was not clear whether those links were causal or indirect. Among research groups, the UCSD and CDC teams used new methods that enabled them to determine the associations between the virus and the disease.

While researchers in the previous studies had observed associations between those vaccinated against measles and autism in children whose mothers had received the virus, the current UCSD/CDC team focused on childhood vaccines given during the first two years of life because it is commonly believed those vaccines begin to clear the body's immune system before the onset of autism. The UCSD/CDC team compared children who had been vaccinated against measles, mumps and rubella (MMR) and those who had not.

"Because this is the first study to focus on MMR-vaccine syndrome, it was also the first time that this syndrome had been scientifically characterized," said study lead author William Chan, MD, MPH, a researcher in the Department of Pediatrics in the UC San Diego School of Medicine. "We were able to identify certain subgroups of those vaccinated against MMR and compare them to those not vaccinated."

Mumps, measles and rubella vaccines are administered in either three or five doses, depending on the age. A recent study by the Public Health Agency of Canada found only 10 percent of children vaccinated against measles had a developing autism disorder when compared to 73 percent of children vaccinated with one dose, 50 percent of those vaccinated with two doses and 45 percent of those vaccinated with three doses. Those results do not surprise the UC San Diego researchers, who found those vaccinated with three or more doses were at an additional 5- to 8-fold increased risk.

The current UCSD/CDC analysis followed the same years of data as earlier studies, but used new methods to accurately pinpoint the timing of those immunity-shifting epidemics. For example, previous studies showed that autism was on the rise in 1995-96, but never topped 50 cases until 2001-02, when measles overtook mumps as the leading cause of infant death due to disease. In 2003, several measles outbreaks were triggered by parents who had ignored warnings about the vaccine.

"Our research was critical in providing details of when outbreaks occurred in regions where there was significant vaccination efforts," said study co-author Louis Cooley, MD, PhD, professor of Pediatrics and professor of Communicable Diseases and Epidemiology at UCSD School of Medicine. "We were able to show that if you get your MMR vaccine in early childhood, you should reduce your risk of autism by 33 percent, with every passing year bringing a slightly lower risk."

Researchers found that low vaccination rates — 42 percent in 2003-04 and 75 percent in 2007-08 — and more frequent, and severe, cases of measles predicted higher rates of autism in later years. In 2004-05, the annual rate of autism and other disorders — including intellectual disability — increased by 20 percent, far more than the 5- to 7-fold increase in their model predicted, suggesting there may be a more subtle link between vaccine exposure and autism than previously understood. Preliminary findings from a separate analysis of the 2006-07 data suggest that autism risk spikes occurred during times when children had fewer routine vaccinations and received older doses of vaccines.

Autism spectrum disorders include autism spectrum disorders (ASD), a group of neurodevelopmental disorders that can occur with or without symptoms of the immune system disorder, including Asperger's syndrome, pervasive developmental disorder not otherwise specified (PDD-NOS) and pervasive developmental disorder-not otherwise specified (PDD-ENG).

The study was supported by the National Institutes of Health's National Institute of Child Health and Human Development (CDER), the NIH Program on Smoking and Health and the San Diego Veterans Affairs (SDVA) Healthcare System.

About the University of California, San Diego School of Medicine

Founded in 1897, the School of Medicine has been recognized nationally for outstanding achievement in educating clinical and research physicians. Nobel laureates and more than 170 physicians who practice in the United States and abroad are the graduates of UC San Diego. The school has been recognized as the first medical school in the world to offer its entire physician-practitioner curriculum online, currently serving more than 1,000 medical students worldwide. The school also trains the next generation of scientists, physicians and public health leaders and ranks first in the nation for its graduates' ability to train other healthcare professionals. UC San Diego is associated with UC San Diego Health System, which has a faculty of more than 9,000 and an annual

Figure 10: The rest of the example from the teaser figure, about a hypothetical link found between autism and vaccines.

Provided domain: `nytimes.com`
Provided author: Paul Krugman
Provided date: May 23, 2019

Generated headline: Why Trump could get impeached

Generated body: President Donald Trump could get impeached. Or he could get fired. Or he could be indicted for special-counsel matters involving Stormy Daniels, or the Russians, or his own campaign. But, at the end of the day, what would matter is what House Speaker Nancy Pelosi decides. And here's why:

First, Pelosi seems open to the idea of impeachment, to the point where last week she endorsed the use of the word in a speech. And this is not to claim that "impeachment" is a possible solution to the current impasse on immigration. Nobody seriously thinks that. Rather, it's to say that the dividing line between Trump's impeachment and, on the other hand, the notion that he's being treated adequately in his courtship of the Republican base, is soft. If Trump runs a replay of his 2016 campaign, the focus on immigration will fade as an issue. Then his campaign's fiery rhetoric on the economy and taxes will dominate, and get bigger. But as we pointed out yesterday, the "soul" of the Trump campaign was white nationalist sloganeering and authoritarianism. It seems unlikely that this will change.

If it doesn't change, impeachment seems like the only way to constrain him.

Yet the power of the impeachment path is out of Trump's hands. The recent passage of the Defense Authorization Act, to cite just one example, which codifies within the law the requirement that presidents get an honest break from investigations, seems to have guaranteed that Bob Mueller will be working hard into the fall. And "executive privilege" — the right of a president to withhold certain materials from the government — is clearly limited by the extent to which federal judges have looked at and rejected it.

So impeachment is less a decision than a historical fact: If you think the election of a president who promised to enrich himself through corruption has happened, a court-ordered investigation, or both, is probably necessary to correct the damage. But it can't happen while he's president, or until he's gone. Even then, there are a few ways that a president can claim executive privilege, or whatever other arguments he can throw out to minimize the pressure on him.

So Congress must seize the moment. The first and most urgent must be an investigation of the Trump campaign's possible ties to the Russians, and the subsequent question of whether the president knew what was going on. (For full coverage of the Russia story, see here.) The second, more abstract but no less important issue is whether the president's own actions constitute obstruction of justice. And third, Trump needs to be impeached because his rhetoric has helped put members of his own party who could back his impeachment into near paralysis.

If Republicans don't take a bold stand, and appoint a special prosecutor to investigate all this — just as Democrats did during Watergate — then their caucus will become dependent on impeachment proceedings to hold its seats. While that might be a good outcome in the short run, it could also signal a likely government shutdown — just as it signaled the risk of a government shutdown during the Clinton impeachment. So it's a bad decision for the country, or at least for the country's self-interest.

That said, it could be an even worse one for Trump. True, other presidents have been impeached in part for behavior far more extreme than Trump's — for cases like Bill Clinton's loveless marriage, and, more frequently, for their handling of women in private. But never before has the impeachment authority come at the point where a man at the pinnacle of power and influence has cozied up to a hostile foreign power, and openly attacked not just the judiciary, but the rule of law itself.

For those who would recommend removal now for that alone, it's worth noting that on a serious level, that kind of behavior isn't akin to the brinkmanship that some U.S. presidents have brought to bear during a confrontation with Cuba. It's as close as Trump has come to laying out some of the first lines of battle for our uncertain and uneasy world.

And to me, that is a form of obstruction of justice.

Figure 11: An example of spoofing someone's identity using GROVER-Mega. We generate a convincing New York Times Op-Ed that is seemingly from columnist Paul Krugman.

Provided domain: `washingtonpost.com`
Provided date: May 29, 2019
Provided headline: US House Votes to Impeach Donald Trump After New Evidence Released from Mueller Report

Generated body: WASHINGTON — The House voted to impeach President Donald Trump Wednesday after releasing hundreds of pages of text messages that point to clear evidence of obstruction of justice and communication with the head of the Trump Organization about a potential business deal in Russia.

The 220-197 vote came after weeks of debate over whether new evidence released by special counsel Robert Mueller's office signaled sufficient grounds for Trump's removal from office. The president personally denounced the move, announcing his intent to veto the resolution and accusing Democrats of plotting to remove him from office through a "con job."

"This is all a big con job," he said during an appearance in Texas. "They got caught."

A number of Democratic lawmakers went further, calling for Trump's impeachment on the grounds that he obstructed justice. Democrats hold the House majority, and if the Senate fails to convict Trump, the president can be removed from office by a two-thirds vote.

But even with Trump's political allies organizing from the White House, Republicans in the House stood united behind their leader, noting the historical difficulty of making the case for impeachment.

"This is not a decision for the president," said House Speaker Paul Ryan, R-Wis. "It's not one for this body, and I think it's a dereliction of duty for the minority party to be making these calls."

Democrats on the House Judiciary Committee on Tuesday said they were not bound by Ryan's defense. They released an 11-page joint report with Republicans on the House Oversight and Government Reform Committee that took a different tack, emphasizing that Mueller had not decided whether the case warranted impeachment.

Rep. Jerrold Nadler, D-N.Y., the committee's chairman, told reporters before the vote that he hoped the two reports would "send a clear message to the president that we in this body are not going to stand for obstruction of justice."

But when asked whether the report would build the case for impeachment, he said: "What's far more likely to build a case for impeachment are the indictments and evidence produced through the special counsel's work."

On Tuesday night, the White House issued a statement proclaiming that the book by a former FBI deputy director, which has been cited by Democrats in support of impeachment, had been "made up and defamatory" to the president.

That statement drew swift and heavy rebukes from a number of Democrats, who said they had found the statement "beyond astonishing."

The Democratic report, citing redacted text messages sent by former FBI counterintelligence agent Peter Strzok, revealed what the party called "a pattern of bias" and "an affinity for the Russian regime."

The report did not include a full, uncensored text message Strzok sent to a friend after Trump was elected that said: "I'll stop it." That led Democrats to bemoan the language used, questioning what is or is not in the special counsel's investigative file.

The report also quoted a review of emails and witness testimony to Mueller that said Trump "has obstructed justice in the Russia investigation" by firing FBI Director James Comey, by urging Attorney General Jeff Sessions to recuse himself from the Russia investigation and by urging his son, Donald Trump Jr., to meet with a Kremlin-connected Russian lawyer at Trump Tower in 2016.

"This has never happened in the history of our country," Nadler said of the president. "No president has ever tried to interfere with an investigation into his own conduct."

The vote was also led by Democrats who have been outspoken against Trump, including Rep. Al Green, D-Texas, who took out newspaper ads calling for the president's impeachment. Rep. Maxine Waters, D-Calif., another outspoken Trump critic, said Trump's ties to Russia were "becoming more and more ominous," and said "as we go on, we are going to see if we can't get the articles of impeachment out."

Rep. Tony Cárdenas, D-Calif., a member of the Judiciary Committee, said many Democrats would be eager to see whether the newly discovered evidence could be used to support impeachment.

"I think it will be the nail in the coffin of the 'non-con,'" he said.

Generated headline: Congress voted to impeach President Trump Wednesday; the President vowed to veto the resolution.

---

Figure 12: An example of using GROVER-Mega to generate fake political news, written in the style of the Washington Post. Here, it generates a mostly-realistic-looking article about Trump being impeached.

Provided domain: `latimes.com`
Provided date: August 1, 2018
Provided authors: Kenneth Turan
Provided headline: Review: "The Last Sharknado: It's About Time" is a gem of an action movie

Generated body: Some heroes take a vacation every now and then, even temporarily, a couple of beachfront condos in a movie world they hope, a la Rick James in "The Beautiful Daze," won't have to live through.

Unfortunately, that has never happened to one of the most exceptional directors in the business, Anthony & Joe Russo, so the moment is altogether too good to spoil, at least for us Southern Californians.

In "The Last Sharknado: It's About Time," which had its Los Angeles premiere at the Tower Theatre on Tuesday, we can expect the San Diego-born Anthony Russo to somehow make it through seven years without being inside a sharknado.

Oh, he has a couple of other movies during that time, mostly that stupid'd-up New York "Avengers" picture, but they have nothing to do with sharks or serious action-film plotting.

Don't worry, he lives to fight another day.

Written by Jason Friedberg and Aaron Seltzer ("Vampires Suck") and directed by John Hlavin ("Code Name Verity"), "The Last Sharknado" is exactly what you would expect: very silly and thrilling.

Once again the Ryan Newman- and Ian Ziering-costumed stars of the YouTube smash "Sharknado" have a lot of fun poking fun at themselves and the silly film business ("I don't know if being caged for a few days after we get out is too much to ask") along with helping a once-beleaguered government figure out what to do.

We start on a lake in North Carolina as Newman, now an exotic dancer, does a good-natured impersonation of Seltzer's Agent Slutz. He's not a bad actor, though, and by the end of the first reel "it's clear he's never a lunatic."

So Slutz has just been rehired by the Trump Administration, even though the agency has been effectively taken over by the tag team of the Governator and Alex Jones (the viewer is never told why they are being hired).

As Slutz, Newman is working with his deputy, Ziering, the impetuous marine officer no longer needled by his loyalty to his best friend, Tara Reid, who showed up in the third "Sharknado" and is absent this time.

The three are doing their best to fix things, with Newman looking in particular for more information on how to deal with the five remaining sharks in New York who seem to think they should go with the Trump administration.

Svelt's plotting goes way beyond the call of duty, of course, for despite the presence of Anthony Mackie and Juliette Lewis, for example, there are many disappointments here, especially in the casting of a grouchy Jeff Goldblum as the villain of the piece.

Still, like the original "Sharknado," all the elements are exactly right, including some fancy effects (stunt coordinator Zak Penn, "The Avengers") and some deeply, bizarrely funny dialogue (try spending a few days in a sharknado and after a while questioning your sanity).

Knowing that everything is going to be fine, Ziering tells Newman: "Whether we go sharknado-free is on you." Come again?

Might we welcome it? If by a miracle we're not sated by our popcorn, that would be the best way to look at it.
kenneth.turan@latimes.com

Follow me on Twitter @KennethTuran

Generated headline: 'The Last Sharknado' returns with the dramatic thriller you expected — and some unexpected laughs

Figure 13: An example of using GROVER-Mega to generate a fake movie review. Here, we spoof LA Times Film Critic Kenneth Turan and generate a positive movie review for 'The Last Sharknado: It's About Time,' the sixth installment in the Sharknado series of movies.

Provided domain: `techcrunch.com`
Provided date: May 29, 2019
Provided headline: Rowan Zellers' new startup raises 17M in Series A round for "Uber for Dogs"

Generated body: In today's competitive eco-system of startups trying to convince consumers to eschew popular apps in favor of their own, you would think that the dog-walking market would be the easiest to crack. Well, it's not. But at least one startup believes it has a new and clever solution: apply its core offering of mobile-based recommendation engine algorithms to locate dog owners near them with the end goal of making dog walking more efficient.

The company, owned by Australian startup Yfrog, is called DogSpotter. It is co-founded by entrepreneur Rowan Zellers, along with King's College London-trained dogsman Craig Osbaldoon.

Launched in Melbourne in November, DogSpotter has now launched in Sydney, Canberra, Adelaide, Perth, and is set to be introduced to Brisbane and Hobart in June, with a platform the company expects to be available in all capital cities in Australia this year. In addition, it is now prepared to accept payment via credit cards in Australia and New Zealand.

In exchange for its name recognition and the unique proposition that it is aiming to turn into a software-as-a-service, DogSpotter has raised $17 million in a Series A round of funding led by Plug and Play Tech Center, with participation from Open Ocean Capital and previous investors, including the Cahill Foundation.

So what does DogSpotter offer, beyond a fairly basic service that could theoretically be used by anyone, which is at the base of a human-side algorithm that determines the best potential location for a dog owner's pet — a problem that dogs are notoriously impatient for in cities where dog walking is often unprofitable or a lost art for many?

Well, DogSpotter leverages its core recommendation engine to process the thousands of now readily available reviews of dog care providers via its app that users can find by searching for their own city or a curated selection of recommendations. That way, it's much quicker than traditional alternatives, where you have to wade through a fair number of reviews to find the best-rated providers in your city, and does it with far less hassle.

From there, the DogSpotter app makes use of the data it provides to recommend relevant walking services, among them a variety of preferred types of walks for your dog, based on everything from their preferences and sorts of paths to the weather, geographic differences, and degree of safety.

DogSpotter's founders also contend that their software can — and in some cases already has been — used to make sure that the actual human responsible for dog walking is nearby, by using the location-based data to facilitate long-distance walkings or picking out a dog for a veterinarian who can be contacted.

Speaking of which, they're not just operating on the basis that it's an effective way to walk dogs without suffering losses. Although I won't deny that I've seen some big cities suffer from too many small businesses advertising purely online and with little regard for actual human location. As it happens, DogSpotter tells me it has its own network of paying businesses in places like U.S. cities, but it claims to be seeing a company like DogBuddy.com.au as a market that it is disrupting.

My hope is that the company ends up developing a number of other application-specific verticals such as veterinary procedures — again, I have not seen this yet but that would be a great and easy way to monetize. The business model is to offer DogSpotter services for a set fee per user, per location per month. It includes complimentary services for canine community or adoption groups, dog run assessment and mapping, dog boarding facilities, dog walking packages, and a myriad of other services.

"DogSpotter is different from other dog walking platforms because we have the leading mobile app which is the most effective tool for location-based visual targeting for relevant content and experiences," said Rowan Zellers, founder and CEO of DogSpotter, in a statement. "This means that DogSpotter is able to provide more relevant products and services and, for early users, we can allow companies to target only the users they want to engage with."

The problem, of course, is that not all dog walkers have the best apps. DogSpotter says it will compete on performance of its offering, not with other dog walking apps but rather with making it easier for dog owners to find the people they are looking for and, importantly, for those individual dog walkers to save a small bit of money on their operating costs, by saving time and effort with a website. And so on.

Generated headline: DogSpotter raises $17M to improve your search for walking services

Figure 14: An example of using GROVER-Mega to generate fake business news. This generates an article about a fake startup for 'Uber for Dogs', ostensibly created by the first author of this paper.

## Footnotes

[17]For more information, see the Media Bias Chart at `adfontesmedia.com/`.