[Reviews · NeurIPS 2019]

Reviewer 1



Detailed comments: Contribution 1 [Significance: High]: New approach for learning and generating multi-field documents i.e. documents where not only a "body" of text is to be generated, but which also contain other fields such as domain, date, authors and headline. The approach is a language modelling approach where these fields are used as context. The approach consists in ordering the fields into two subsets during the training in an attempt to address issues such as expensive marginalization or dealing with a large number of field orderings during inference time. The approach seems novel to me as I am not personally aware of previous work that approaches this problem in this way. Question: It wasn't clear to me if some of the fields have some restricted domains from which they can be sampled. For example, for the authors name, are these being sampled from the general vocabulary or some subset of it that focuses on names? Because it's not evident how the model will abstain from, say, sampling something evidently wrong (e.g. the word chair or bed) for a person's name. If I'm missing something, this needs some clarification. Contribution 2 [Significance: High] New dataset: The authors present a new dataset RealNews which was crawled from Common Crawl. The dataset should be a valuable resource for future research (on generation and classification) around the news domain given its scale and its inclusion of the accompanying metadata. Contribution 3 [Significance: Medium]: Extensive experimentation and evaluation: The authors present extensive experimentation along with many intuitions and observations. Comment 1: One main concern I have is that an overall read of the paper might give the impression of blurring the distinction between the notions of real, fake, generated, human-written. In Section 5, the authors talk about the Verifier being used to classify “human-written” vs “machine-generated”. In Section 2, the Verifier’s “goal is to classify new stories as real or fake”. Indeed, not all machine-generated text is outright “bad” or fake. Ultimately, as NLP researchers, we want to research and design and build stronger generation systems that can be used, e.g., for summarization, translation, answering, etc. I found it disappointing that the paper did not touch on the aspect of classifying articles as “real” or “fake” in the sense of truthful versus not truthful given that the threats of these massive models is not that their output is “fake” in the sense of “machine-generated” as much as it is not truthful. As the authors point out and use as part of their data, many websites are already considered propaganda and so their content is likely to be not truthful or biased. And this is one dimension they could’ve touched upon in their evaluation. Comment 2: Evaluation only touches on aspects related to body of text. However, the model is supposed to generate a full article and so I think the authors should have had any kind of evaluation for metadata (e.g. similarity between headline and body of text, how often the generated author names even make sense and are actual names of people, how often the dates are actually correct dates, etc). Comment 3: While the paper makes the point that Grover is the best at detecting Grover generations, it would’ve been nice to see how Grover does on the generations of other models versus those corresponding models. Contribution 4 [Significance: Low]: The ethical agenda: I think this aspect of the paper is very much appreciated and welcome and needed to foster more discussions around the ethical issues revolving around text generation especially in the era of these recent massive models. I have to admit that the intro led me to have high expectations and I found that this ethical agenda was kinda underwhelming. It's a part of the conclusion and doesn't provide new insight nor does it add much to the existing conversation. For example, using models that work as filters is something that is already being done (to a certain extent) by social media platforms like Facebook or Twitter. Similarly, a lot of the heat that OpenAI received following its decision to not release its full model focused on the fact that their models could be used as classification models that could deter the attacks Clarity: Overall, the paper is written very well. It's well-structured and flows smoothly. Few parts were somewhat difficult to follow e.g. mapping Section 5.2 to Table 1 was confusing, making clear what the work is evaluating (discrepancy between end of Section 2 and Section 5 as highlighted above). Clarification 1: One main concern I have is that an overall read of the paper might give the impression of blurring the distinction between the notions of real, fake, generated, human-written. In Section 5, the authors talk about the Verifier being used to classify “human-written” vs “machine-generated”. In Section 2, the Verifier’s “goal is to classify new stories as real or fake”. Clarification 2: What do you mean by “news domains” (Line 132). Significance: The work is important both because of the problem domain it’s addressing (the spread of fabricated news using neural ML) and the analysis and observations it presents. It’s also important because it’s introducing a new resource that should foster more research on this topic (hopefully, the dataset would be given to any credible team of researchers as the authors do not specify their intention to publicly release the dataset, instead making it available through requests). Typos: Figure 5 in the legend: +Grover-Base Generations* Other comment: This is more of a philosophical comment. I understand authors of papers usually – and rightly so – want to have catchy titles for their paper or their models. In this case however, the authors via their paper –potentially to be published at our field’s flagship conference— would be serving to popularize a concept, namely “fake news”, that was used by the current US administration to not only shut down essentially any criticism of the administration’s policies or actions, but even more dangerously to attack entire groups of media outlets calling them “fake news” and enemy of the people. A quick Google Trends search shows that the expression “fake news” was barely used before 2016 and only spiked after the new administration came in, so that combination of these 2 words was not popular at all before that point in time. The authors already use in their paper alternative expressions such as “neural disinformation” which seems to me rather catchy, and they also use other synonyms for “fake” such as false (others that come to my mind include falsified or fabricated). As such, I encourage the authors to reconsider their choice of words both in their title and in the body of their paper.

Reviewer 2



POS AUTHOR FEEDBACK I thank the authors for their feedback. In particular, the comment around having used BERT in a multi-field setting I believe is an important addition. Unfortunately, the allowed space is not enough to explain how you used BERT for generation. Regarding the annotator guidelines however I am more concerned, as it seems that the questions did not ask about the content specifically, but about the trustworthiness of the source, of which the article is only a proxy. In my opinion this only augments the risk that you are doing style classification, where one source is main-stream media and another are media with strong political biases. To support the claim that GROVER is good in discriminating machine-generated content in general this should be benchmarked with other (generative) language models in all directions. ===== ORIGINAL REVIEW ====== This paper proposes a modification of GPT2 in order to: - guide the textual generation without the need of prompting. It does so by providing "contextual" information in the form of fields - generate multi-field documents In practice, this boils down to generate the field sequentially and generate them left-to-right; plus a carefully tuned training mode that drops out fields randomly. The author show that they are able to generate extremely credible news articles. One of the strongest argument of the paper is the ability of that model to generate more credible news-article than real (human-generated) fake news. They then benchmark different models to detect machine-generated text, concluding that the best model is the one that uses the representation learnt by the same model that generated the text. The paper concludes on some insights on what are the signals to detect machine-generated text and discusses ethical considerations on the release of NLG models, adding to an important discussion in the field There are lots of things in this paper! It is obviously a very timely contribution to an important debate, it is already being discussed in NLG circles and will probably be topic of conversations in the corridors of the convention centre of Vancouver if accepted. The paper is very clearly written, (important) details of training are given in the Appendix and the authors make a good work of condensing many ideas into 8 pages. None of the contributions however is very significant by itself. In particular, with respect to what seems to be presented as the strongest point of the paper: - presentation of a new model: this is a bit over-sold, it is just a modification of GPT2 - "humans find these generations to be more trustworthy than human-written disinformation". This point (Fig 4, last two groups, 2.42 vs 2.19; lines 170-172) is indeed striking. However, it is hard to understand what this means without a complete description of what was asked to the turkers. Was "overall trustworthiness" the only wording given? If so, what exactly means "content sensibility" (I would have a hard time figuring out what this means without more information)? Also, it is not clear why the same trend does not hold for Human-News vs Machine-News. Nothing is said about how many annotators labeled each article and what the inter-annotator agreement was. As the trends in both groups are the same for Style and Overall, one interpretation (which is not discussed) is just that humans get fooled by the style more than by the content. This is, if a human would write a propaganda article in the style of (let's say) the NYTimes then its Overall score would go up. Propaganda websites might not do that because that is not "their" style and what their readers expect. While the generator is conditioned on the source, Fig 4 could well indicate that the style of reputable news article leaks towards propaganda web-sites - Grover generates better news article than GPT2 (Fig 3). Isn't that obvious? Grover is trained on in-domain data, while GPT2 is not. Also, it seems obvious that the meta-data (in particular headline and domain, but also authors) helps to reduce perplexity. This is a good sanity check, but not really impressive. - Grover is better to detect Grover-generated text (Sect 5.2). This seems to say less about Grover itself than about "it takes a thief to catch a thief". This is, I would expect a GPT2 model trained on RealNews to be better to detect generated text with that model than Grover trained on out-of-domain multi-field documents. This conclusion is pointed out in the paper, but not tested beyond Grover. I believe a paper which starts from this as main idea would be a very strong submission. Finally, I believe the choice of sources (Appendix B) is at least controversial. I understand that it is hard to find a good training data for "fake news", but currently the "propaganda sites" mix "web-sites that spread misinformation" and those with "strong political affiliations". For those last one, I am not an expert in US politics, but it seems to be rather "strong political affiliations, *on the right* ". Why are there no extreme-left websites? More-over, this mix of obvious fake-news with other more fringe-news creates biases (both at training and at human evaluation) which are not sufficiently discussed in my opinion.

Reviewer 3



This paper proposes to model fake news generation as a joint distribution over domain, date, headline, body, and author. In order to perform controllable news generation, the authors model the ability to generate any subset of these fields, given the other subset. To avoid modelling all possible permutations of these fields, GROVER assumes a canonical ordering of all the fields, and models generating one random subset given the other subset, in order. At training time, fields in news articles are randomly partitioned into two sets and GROVER maximizes the likelihood of generating one subset given the other. The underlying language model is a large Transformer Model similar to GPT (Radford et al). The model thus trained achieves significant perplexity reduction in generation of news article body when conditioned on metadata. Perhaps quite surprisingly, human annotators rate GROVER generated propaganda higher in terms of style and overall trustworthiness than human written propaganda. The paper then compares techniques for detecting fake news by trying to discriminate GROVER generated articles from real news articles (both from April 2019). The experiments show that GROVER models outperform strong BERT and GPT baselines in this task, perhaps unsuprisingly. The results hold when the generator size is larger than the discriminator size. The difference between performances of GROVER and BERT remain when the fake news articles that are fed to the training algorithm are sampled from smaller GROVER(-base or -large) while the test articles are from GROVER-mega. The paper presents an explanation for why GROVER discriminators have the right inductive bias when compared with BERT. This is because the generation process explicitly takes word probabilities into account (and with nucleus sampling, avoiding generating words from the tail); the disciminator having access to these word probabilities can better model the generation process. Originality 1. The work presents a new technique of incorporating metadata into account for neural text generation. While the process for sampling fields is simple, it is original and useful/ 2. The semi-supervised experiments for neural fake news detection are very well done and exciting. The setting where an adversary has access to a large neural net model and the verifier has access to a small one is quite original and interesting. cons: Minor: The paper uses describes GROVER's discriminative model being the best at identifying GROVER-generated fake news text as "counterintuitive" multiple times. It certainly doesn't help the early discussion and is not discussed as a "counterintuitive" find in the later sections. This finding certainly does not sound very surprising and should not be proposed as such. Quality pros: 1. The techniques used in the paper are mostly clear and represent the latest norms in the community. 2. The studies in section 5.3 and 6 are quite useful in pointing out the effect of nucleus sampling on the accuracy of the discriminator. cons: 1. Even though section 6 is titled "How does a model distinguish between human and machine text?", it actually does not meaningfully explain how the discriminative model distinguishes between human and fake news text specifically with a limited variance version. More below on the clarity of section 6. 3. It is unclear how there is "minimal overlap" between the discriminator and generator model as claimed in 181-183. Also unclear if the discriminator is initialized from generator parameters. Clarity pros: 1. The paper is very clearly written and is appropriately supported by figures and colors. cons: 2. Overall section 6 is a bit of hodgepodge of two ideas: how does GROVER detect real vs neural fake news and what is the inductive bias in GROVER that gives it an advantage over BERT. Disentangling these two ideas will make this section a lot clearer. Significance 1. This is a very important and timely piece of work. The impact of the released model and released dataset is likely to be huge in the community. 2. The line of questioning around fake news detection and various artefacts around sampling distortions around neural text generation are both fertile grounds for future work.

[Author Response · NeurIPS 2019]

We thank the reviewers for their helpful comments. Overall, reviewers said the work was **timely** (R2, R3) and **important**
(R1, R3), with R3 adding: "the impact of the released model and released dataset is likely to be huge in the community."

We are excited that reviewers rated our Grover generator as being a **simple** (R3) yet **novel** (R1, R2) approach for
multi-field document generation. Reviewers noted that Grover generates "extremely credible" articles (R2) and that due
to its inductive bias as a generator (R3) it is also a state-of-the-art detector of neural fake news.

Last, reviewers said that our experiments on neural fake news detection in a semi-supervised setting were extensive
(R1) and "very well done and exciting" (R3), particularly analysis about when the adversary has more resources.

──────────── **Reviewer 1** ────────────
(Contribution 3, Comment 1) / (Clarity Clarification 1): **machine authorship vs. fake?** Good point, we will clarify our
terminology in revision. Just to clarify, Grover is pretrained only on human-written truthful news; propaganda websites
are excluded. We did additional experimentation and found that Grover discriminates between human-written real news
vs. human-written propaganda with **98% accuracy**, and will add this (and discussion) to the revision.

(Contribution 3, Comment 2): **evaluating consistency with metadata?** Good idea: we did additional human evaluation
on the consistency of the article body with the headline, date, and author. We found that **generations are largely**
**consistent** overall. For instance, human propaganda articles are consistent with the headline with an average score of
2.85/3 (higher is better) but machine-written propaganda gets 2.64/3; the news scores are similar.

(Contribution 3, Comment 3): **spotting generations of other models?** We ran additional experiments here: Grover
detects GPT2-generated news (released by OpenAI) with **96% accuracy**, even without finetuning on GPT2 generations.

(Philosophical comment): **fake news term?** We appreciate this point and will revisit the word choice. In our draft,
we followed the lead of the following political science paper, which has an extensive discussion about recommended
terminology: *Lazer, David MJ, et al. "The science of fake news." Science 359.6380 (2018): 1094-1096.*

(Contribution 1, Q): **All fields share the same vocabulary** - we'll clarify this in revision. We haven't seen the model
generate field-inappropriate tokens (like an invalid date), perhaps due to extensive pretraining and nucleus sampling.

──────────── **Reviewer 2** ────────────
**Novelty of Grover over GPT2?** We believe that our "novel way to guide generation" makes Grover novel, not just an
'adaptation'. Indeed, GPT(2), BERT, XLnet, and Grover share the same backbone but learn from different objectives.

**What is given to the turkers?** We will provide the full prompt in revision along with other details (we used 3
annotators) and discussion. For overall trustworthiness for instance, we asked "Does the article read like it comes
from a trustworthy source?" Thus it emphasizes style, while content sensibility measures whether the semantic
content is believable; thereby **disassociating style vs. content**. The results in the paper show that news written in a
propaganda style appears less trustworthy, but the *content* of human and machine propaganda is equally sensible. The
real news→Grover news results likely generalize to propaganda→Grover propaganda, were this desired.

**"It takes a thief to catch a thief"?** We thank the reviewer for this comment and will revise the paper to be more precise.
However, we *did* test this hypothesis beyond Grover. We trained our BERT model on RealNews in a multi-field setting.
Nevertheless, BERT is worse at neural fake news discrimination compared with Grover. We found this surprising
because NLP leaderboards for discriminative tasks, like GLUE, show the dominance of deep bidirectional models like
BERT over unidirectional ones like GPT. Even though Grover cannot handle right-to-left dependencies, it still is a
state-of-the-art discriminator because its inductive bias matches that of a generator's (R3).

**Choice of sources?** We used only those propaganda sites whose strong political affiliations make it so they spread
disinformation, and as rated by the Media Bias Chart (an updated data-driven list of news sources in terms of bias and
truthfulness). This includes extreme-left websites like naturalnews.com. We will clarify this in revision.

──────────── **Reviewer 3** ────────────
**Examples that support the analyses?** We will investigate ways to visualize the discriminator in action, including
color-coding predictability like (Strobelt and Gehrmann, 2019).

**Experiments on human propaganda?** see response to R1 (Contribution 3, Comment 1).

**'counterintuitive'?** Thanks for the comment. We will change the word choice accordingly. See our response to R2 "it
takes a thief to catch a thief" for why we found it interesting.

**Minimal overlap between models?** Good point - the discriminator is initialized from generator parameters at an
earlier stage of training (L182) but measuring overlap between models is challenging. Still, in our paper we consider
discriminating from a larger/smaller generator (so no parameter overlap) and even in this case Grover is the best detector.

**Clarity of section 6?** Thanks for the comment; we will better disentangle these two key ideas in revision.

[Meta-Review · NeurIPS 2019]

This is important and timely work. As the reviewers mention, this is likely to be a "buzzy" paper. The primary objection to the paper is that it risks slipping between "fake text" (generated by computers but with no judgements about truth) and "fake news" (untrue statements written by humans or computers). There's an important difference between an auto-generated summary of a basketball game that reports accurate events and a hostile article written to misinform. I'd like to see more discussion in the paper about what the systems are actually identifying. Currently I (and R2) find it difficult to interpret what is being distinguished and why. I'm more optimistic than R2 about the feasibility of adding some results on adding other generators like BERT.